

# Socioeconomic status is not associated with health-related quality of life in a group of overweight middle-aged men

José G.B. Derraik[1,2,3], Benjamin B. Albert[1], Martin de Bock[1,4], Éadaoin M. Butler[1,2], Paul L. Hofman[1] and Wayne S. Cutfield[1,2]

[1] Liggins Institute, University of Auckland, Auckland, New Zealand
[2] A Better Start—National Science Challenge, Auckland, New Zealand
[3] Department of Women's and Children's Health, Uppsala Universitet, Uppsala, Sweden
[4] Department of Paediatrics, University of Otago, Christchurch, New Zealand

## ABSTRACT

Socioeconomic status is a known determinant of health. In secondary data analyses, we assessed whether socioeconomic status affected health-related quality of life in a group of overweight (body mass index 25–30 kg/m$^2$) middle-aged (45.9 $\pm$ 5.4 years) men, recruited in Auckland (New Zealand). Health-related quality of life was assessed with SF-36v2 three times: at baseline, and 12 and 30 weeks later. Socioeconomic status was determined by geo-coded deprivation scores derived from current address using the New Zealand Index of Deprivation 2006 (NZDep2006), as well as capital value of residence. Univariable and multivariable analyses showed no associations between measures of socioeconomic status and any mental or physical health domains. Our findings may reflect the fact that these men are not currently experiencing comorbidities associated with overweight.

## INTRODUCTION

Socioeconomic status (SES) is a major determinant of health. Across the socioeconomic spectrum there are stepwise improvements in mortality and morbidity with increasing wealth (*Adler & Ostrove, 1999*; *Braveman & Gottlieb, 2014*). SES is likely to affect health through complex direct and indirect pathways (*Adler & Newman, 2002*; *Braveman & Gottlieb, 2014*). For example, people with lower incomes report greater financial obstacles to effective treatment, often leading to delayed diagnosis and suboptimal management of health conditions (*Osborn et al., 2016*).

Health-related quality of life (HRQL) refers to the effects of health, illness and treatment on perceived quality of life (*Ferrans et al., 2005*). SES has previously been described to affect HRQL in a nationally representative cohort of Canadian adults (*Ross et al., 2012*), in adults with a chronic disease in Germany (*Mielck, Vogelmann & Leidl, 2014*), and among males (but not females) in Japan (*Yamazaki, Fukuhara & Suzukamo, 2005*). Among patients with rheumatoid arthritis and other chronic conditions, there was evidence that lower income

Corresponding author
José G.B. Derraik,
j.derraik@auckland.ac.nz

was associated with poorer HRQL (*Alishiri et al., 2008*; *Ovayolu, Ovayolu & Karadag, 2011*), but females made up the vast majority of both study populations. Of note, in the USA, disparities in self-rated health when stratified by income group were widest in adults aged 45 to 54 years (*Robert et al., 2009*). Previous studies have also indicated that poorer SES is associated with adverse metabolic outcomes, but two of these studies observed this association only among women (*Chichlowska et al., 2008*; *Lim et al., 2012*), with a third investigation showing only a week association among male participants (*Loucks et al., 2007*).

We are not aware of any previous study in New Zealand into the association between SES and HRQL in middle-aged males. Importantly, the prevalence of obesity among adult males in New Zealand increased from 26% in 2006/07 to 30.5% in 2015/16, and it was highest in the most deprived areas at 40.7% (*Ministry of Health, 2016*). Thus, in light of previous evidence and given the increasing prevalence of obesity in New Zealand men (and its particularly high prevalence in areas of low socioeconomic status), we aimed to investigate whether lower SES would also be associated with HRQL in a phenotypically homogenous group of overweight middle-aged men.

## METHODS

### Participants

This study consisted of secondary analyses of data from a 30-week randomized crossover trial in Auckland (New Zealand) assessing the effects of supplementation with olive leaf extract on insulin sensitivity (*De Bock et al., 2013*). Participants were middle-aged men (35–55 years) who were overweight (body mass index 25–29.99 kg/m$^2$). Key exclusion criteria in the original trial were tobacco smoking, use of illicit drugs, diabetes mellitus, or taking any medications that could affect insulin sensitivity. In the current study, participants with incomplete HQRL data were also excluded.

### Assessments

SES was determined for each participant by geo-coded deprivation scores derived from current address using the New Zealand Index of Deprivation 2006 (NZDep2006) (*Salmond, Crampton & Atkinson, 2007*). This index is based on household census data reflecting nine aspects of material and social deprivation to divide New Zealand into tenths (scored 1–10) by residential address (*Salmond, Crampton & Atkinson, 2007*). Scores of 1 represent the least deprived areas and 10 the most deprived ones (*Salmond, Crampton & Atkinson, 2007*). Scores are derived from units covering a small area, each reflecting approximately 87 people (*Salmond, Crampton & Atkinson, 2007*). Although NZDep2006 scores apply to areas rather than individual people, they are considered reasonable indicators of SES where in depth individual measures are unavailable (*Salmond & Crampton, 2012*). In addition, the capital value of each participant's residence was obtained from publicly available data from the Auckland City Council.

Each participant had HRQL assessed at the start of the trial and then at 12 and 30 weeks, in order to provide a more robust estimate of well-being for each individual. HRQL was assessed with the validated New Zealand/Australia adaptation of the SF-36v2 Health

Survey (*Frieling, Davis & Chiang, 2013*). The SF-36v2 is based on subjective measures of well-being, measuring perception of health covering well-being, functional status, and overall evaluation of health (*Ware Jr, 2000*). The SF-36v2 assesses four physical health domains (bodily pain, physical functioning, role limitations related to physical problems, and general health) and four mental health domains (social functioning, vitality, role limitations related to emotional problems, and general mental health). Each individual domain is made up by a number of specific items, which have been described and discussed in detail by *Ware Jr (2000)*. Physical and mental health domains were also quantified using summarizing scores.

Weight and height were measured during clinical assessments, and body mass index (BMI) calculated. The long format of the International Physical Activity Questionnaire (IPAQ) was used to assess physical activity levels (*Hagstromer, Oja & Sjostrom, 2006*). The questionnaire was self-administered and was completed during clinical assessment, reporting on the participant's physical activity levels over the previous 7 days. The IPAQ covers four domains of physical activity: work-related, transportation, housework/gardening, and leisure time. Its validity has been shown against accelerometer data, so that it has been used widely, across a diverse range of populations.

Nutritional intake was evaluated using 3-day dietary records that were collected at each clinical visit. These records contained itemized descriptions of dietary intake during one day in the weekend and two week days (i.e., Monday to Friday). Nutritional intake was recorded using standard household measures and information obtained from food labels if necessary. A single trained investigator instructed all participants and also reviewed all records, to correct any errors or omissions, and clarify any unclear entries. The same investigator entered all records into the Foodworks software (v6.0, Xyris Software, Brisbane, Australia).

## Statistical analysis

Pearson's correlation coefficients or Spearman's rank correlations were initially run. Associations between NZDep2006 scores and SF-36v2 outcomes were examined using general linear mixed models based on repeated measures. Models adjusted for randomization sequence, timing of assessment, and on-going use of cholesterol-lowering and/or antihypertensive medications, as well as participant's age, BMI, and physical activity level (IPAQ score).

Apart from continuous associations, stratified analyses were performed splitting the group in half according to the levels of socioeconomic deprivation: Lower SES (NZDep2006 scores 4 to 10) and Higher SES (scores 1 to 3). Baseline differences between groups were examined using t-tests, Chi-square tests, or Fisher's exact tests, as appropriate.

Identical continuous and stratified analyses were also run for capital value of residence. The two groups for the stratified analyses were: Lower capital value (NZ $230,000 to 590,000) and Higher capital value ($\geq$NZ $600,000).

Statistical analyses were performed using Minitab v.16 (Pennsylvania State University, State College, PA, USA) and SAS v.9.3 (SAS Institute Inc. Cary, NC, USA). Where appropriate, outcomes were log-transformed to approximate a normal distribution prior
to analyses. Outcome data are presented as estimated marginal means adjusted for the confounding factors in multivariable models with respective 95% confidence intervals (back-transformed for logged data). All tests were two-sided, with statistical significance maintained at $p < 0.05$, without adjustments for multiple comparisons.

### Ethics

The original trial was registered with the Australian New Zealand Clinical Trials Registry (#336317), with ethics approval granted by the Northern Y Regional Ethics Committee (NTY/11/02/015). All participants provided written informed consent.

## RESULTS

Forty-five subjects participated in the clinical trial (*De Bock et al., 2013*), but seven had incomplete HQRL data and were excluded. Thus, we studied 38 overweight men (BMI 27.3 $\pm 1.4\,\text{kg/m}^2$) aged 45.9 $\pm 5.4$ years (range 34.5–55.6 years), who were mostly of New Zealand European ethnicity (89%). Three participants were on antihypertensive medication, three were on lipid-lowering medications, and two participants were on both. No participants had any other physical or mental health co-morbidities.

Comparisons between our group of participants and New Zealand normative data have been previously reported (*Derraik et al., 2014*). Briefly, our cohort had similar scores in all mental health domains when compared to normative data, but our participants displayed better role physical ($p < 0.001$), physical functioning ($p < 0.001$), bodily pain ($p = 0.012$), and general health ($p = 0.009$) scores (*Derraik et al., 2014*).

### NZDep2006

NZDep2006 scores were not correlated with participants' scores in any mental or physical health domain. Multivariate models examining linear associations also yielded negative results. Subsequent stratified analyses corroborated these negative findings. Participants of lower SES displayed similar scores across all mental and physical health domains as participants of a higher SES (Table 1). This lack of association was observed even when models adjusted for important confounding factors (Table 1). Note that there were also no differences in demography, clinical parameters, dietary intake, or physical activity levels between SES groups (Table 1).

### Capital value of residence

Information on capital value was available for 35 of 38 participants. Men living in residences of a higher capital value were 4.2 years older on average ($p = 0.020$), but had similar clinical characteristics at baseline, physical activity levels, and dietary intake (Table 2). Similarly to NZDep2006, there were no significant correlations between residential capital value and any mental or physical health domains, with multivariate models providing similar results. Stratified analyses yielded no significant differences in physical or mental health domains between groups separated according to capital value (Table 2).

**Table 1  Health-related quality of life (HRQL) data on 38 middle-aged overweight men according to socioeconomic status as per NZDep2006 scores.** Each participant was evaluated three times over a 30-week period. Baseline data are means ± standard deviations or $n$(%). Physical health and mental health data are estimated marginal means and respective 95% confidence intervals from general linear mixed models based on repeated measures, adjusted for randomization sequence, timing of assessment, and on-going use of cholesterol-lowering and/or antihypertensive medications, as well as participant's age, BMI, and physical activity level (IPAQ score). Lifestyle data are estimated marginal means and respective 95% confidence intervals from general linear mixed models based on repeated measures. Note that higher physical and mental health scores represent better outcomes; lower NZDep2006 indicate lower levels of socioeconomic deprivation (i.e., wealthier status).

| | | Lower socioeconomic status | Higher socioeconomic status | $p$-value |
|---|---|---|---|---|
| **n** | | 18 | 20 | |
| **Baseline demography** | NZDep2006 | 6.2 ± 2.1 | 2.4 ± 0.8 | <0.0001 |
| | Capital value of residence (NZ$) | 565,313 ± 164,747 | 782,368 ± 353,767 | 0.025 |
| | Age (years) | 45.1 ± 5.7 | 46.7 ± 5.2 | 0.37 |
| **Baseline clinical data** | BMI (kg/m$^2$) | 27.4 ± 1.5 | 27.6 ± 1.3 | 0.71 |
| | Taking cholesterol-lowering medication | 4 (22%) | 1 (5%) | 0.17 |
| | Taking antihypertensive medication | 4 (22%) | 1 (5%) | 0.17 |
| | Systolic blood pressure (mmHg) | 125.2 ± 8.4 | 124.5 ± 11.6 | 0.84 |
| | Diastolic blood pressure (mmHg) | 78.4 ± 6.0 | 78.1 ± 7.3 | 0.88 |
| **Lifestyle** | Physical activity levels (IPAQ score) | 2,117 (1,329–3,372) | 1,902 (1,226–2,951) | 0.73 |
| | Total energy intake per day (kJ) | 9,428 (8,576–10,279) | 9,222 (8,381–10,064) | 0.73 |
| | Energy from saturated fat (%) | 13.2 (12.2–14.2) | 12.5 (11.5–13.5) | 0.31 |
| | Energy from sugar (%) | 15.5 (13.5–17.5) | 16.9 (14.9–18.9) | 0.29 |
| | Fibre intake per day (g) | 23.3 (20.4–26.7) | 23.3 (20.3–26.6) | 0.98 |
| | Consumed any alcohol | 16 (89%) | 17 (85%) | 0.99 |
| | Energy from alcohol (%) | 6.5 (4.1–8.9) | 5.5 (3.1–7.8) | 0.53 |
| **HRQL** | **Physical health** | | | |
| | Physical component summary | 56.4 (54.9–57.9) | 57.4 (55.5–59.3) | 0.32 |
| | General health | 76.1 (71.0–81.3) | 78.3 (72.1–84.5) | 0.56 |
| | Physical functioning | 95.4 (92.4–98.5) | 93.3 (89.6–97.0) | 0.37 |
| | Role limitations due to physical problems | 96.0 (93.6–98.5) | 95.0 (92.1–98.0) | 0.58 |
| | Bodily pain | 80.6 (74.9–86.4) | 82.5 (75.5–89.4) | 0.66 |
| | **Mental health** | | | |
| | Mental component summary | 52.2 (49.3–55.1) | 50.3 (46.8–53.8) | 0.36 |
| | Mental health | 78.3 (73.3–83.4) | 75.1 (69.0–81.3) | 0.38 |
| | Vitality | 65.8 (60.1–71.6) | 63.4 (56.4–70.3) | 0.55 |
| | Social functioning | 92.0 (85.2–98.7) | 90.1 (81.9–98.3) | 0.70 |
| | Role limitations due to emotional problems | 92.2 (87.2–97.1) | 88.6 (52.6–94.6) | 0.32 |

## DISCUSSION

We did not observe any associations between SES and HRQL in our group of overweight middle-aged men, whether assessed by NZDep2006 or capital value of residence. Our findings contrast to the results of *Minet Kinge & Morris (2010)* who observed lower HRQL scores among obese and overweight individuals with a lower SES than those of the same weight with a higher SES. Studies in Canada (*Ross et al., 2012*) and Greece (*Pappa et al., 2009*) also observed that greater affluence was associated with higher HRQL.
**Table 2  Health-related quality of life (HRQL) data on 35 middle-aged overweight men according to the capital value of their residential address.** Each participant was evaluated three times over a 30-week period. Baseline data are means ± standard deviations or $n$(%). Physical health and mental health data are estimated marginal means and respective 95% confidence intervals from general linear mixed models based on repeated measures, adjusted for randomization sequence, timing of assessment, and on-going use of cholesterol-lowering and/or antihypertensive medications, as well as participant's age, BMI, and physical activity level (IPAQ score). Lifestyle data are estimated marginal means and respective 95% confidence intervals from general linear mixed models based on repeated measures. Note that higher physical and mental health scores represent better outcomes; lower NZDep2006 indicate lower levels of socioeconomic deprivation (i.e., wealthier status).

| | | Lower capital value | Higher capital value | *p*-value |
|---|---|---|---|---|
| *n* | | 17 | 18 | |
| **Baseline demography** | Capital value of residence (NZ$) | 450,882 ± 93,261 | 902,500 ± 259,718 | <0.0001 |
| | NZDep2006 | 4.7 ± 2.9 | 3.6 ± 1.9 | 0.20 |
| | Age (years) | 43.6 ± 5.0 | 47.8 ± 5.3 | 0.020 |
| **Baseline clinical data** | BMI (kg/m$^2$) | 27.7 ± 1.4 | 27.4 ± 1.4 | 0.58 |
| | Taking cholesterol-lowering medication | 2 (12%) | 3 (17%) | 0.99 |
| | Taking antihypertensive medication | 2 (12%) | 3 (17%) | 0.99 |
| | Systolic blood pressure (mmHg) | 124.8 ± 8.0 | 125.8 ± 12.4 | 0.78 |
| | Diastolic blood pressure (mmHg) | 78.6 ± 6.1 | 78.9 ± 7.0 | 0.88 |
| **Lifestyle** | Physical activity levels (IPAQ score) | 2,216 (1,373–3,577) | 1,866 (1,174–2,964) | 0.60 |
| | Total energy intake per day (kJ) | 8,756 (8,073–9,439) | 9,243 (8,532–9,954) | 0.32 |
| | Energy from saturated fat (%) | 13.6 (12.6–14.6) | 12.4 (11.4–13.4) | 0.10 |
| | Energy from sugar (%) | 15.9 (13.7–18.0) | 16.5 (14.3–18.7) | 0.65 |
| | Fibre intake per day (g) | 22.8 (19.9–26.2) | 22.8 (19.8–26.3) | 0.99 |
| | Consumed any alcohol | 14 (82%) | 16 (89%) | 0.99 |
| | Energy from alcohol (%) | 5.7 (3.2–8.3) | 5.8 (3.4–8.1) | 0.99 |
| **HRQL** | **Physical health** | | | |
| | Physical component summary | 57.7 (56.0–59.4) | 55.8 (54.1–57.5) | 0.06 |
| | General health | 77.0 (70.9–83.0) | 76.6 (70.7–82.6) | 0.93 |
| | Physical functioning | 95.0 (91.3–98.7) | 94.3 (90.7–97.9) | 0.83 |
| | Role limitations due to physical problems | 96.9 (94.1–99.7) | 94.7 (92.0–97.3) | 0.22 |
| | Bodily pain | 85.7 (78.1–93.3) | 77.6 (70.2–85.0) | 0.11 |
| | **Mental health** | | | |
| | Mental component summary | 50.8 (47.4–54.2) | 52.4 (49.1–55.8) | 0.46 |
| | Mental health | 77.0 (71.1–83.0) | 77.5 (71.7–83.4) | 0.90 |
| | Vitality | 63.4 (56.7–70.1) | 66.3 (59.7–72.9) | 0.51 |
| | Social functioning | 91.3 (83.6–99.0) | 92.4 (84.8–99.9) | 0.83 |
| | Role limitations due to emotional problems | 89.1 (83.6–94.6) | 93.2 (87.8–98.5) | 0.26 |

However, the existing evidence for the association between HRQL and SES is not consistent. In Japan, *Yamazaki, Fukuhara & Suzukamo (2005)* observed a strong association between income and HRQL amongst Japanese men but not women, while an older study in that country found a very modest effect of income on HRQL (*Asada & Ohkusa, 2004*). *Pappa et al. (2015)* observed that among Roma adults in Greece material deprivation was associated with lower HRQL scores on certain domains (general health and vitality), but with higher HRQL scores on others (role physical and role emotional), findings that the authors were unable to explain. Other studies focusing on physical health (more specifically

on prevalence and characteristics of the metabolic syndrome) also failed to observe any association with HRQL among men in the United States (*Chichlowska et al., 2008*) and South Korea (*Lim et al., 2012*). It is possible that some of these inconsistencies may be associated with for example, varying levels of education, as previous studies have shown a strong relationship between education levels and HRQL (*Devlin, Hansen & Herbison, 2000*; *Lacey & Walters, 2003*; *Mielck, Vogelmann & Leidl, 2014*). Of note, *Miravitlles et al. (2011)* observed that lower levels of education were associated with lower HRQL, but no association was observed for occupational status. Thus, it is possible that while we did not observe an association between area-level deprivation or capital value of residence with HRQL, the results might have been different if it had been possible to account for the participants' levels of education.

Nonetheless, while SES did not appear to affect HRQL in our participants at the time of assessment, it is unlikely to remain unimportant over time. It has been proposed that people of lower SES face a 'double burden', namely increased health complications as well as lower HRQL when their health declines (*Mielck, Vogelmann & Leidl, 2014*). Participants in our study were middle-aged, overweight but not obese, did not smoke, and had no significant comorbid conditions. Over time, these men may put on more weight, and adult weight gain is associated with increased risk of chronic diseases in men (*De Mutsert et al., 2014*). This progression from overweight to a diseased state is more likely to occur in those of lower SES. Thus, even though our results covered a period of approximately 30 weeks, our findings may not reflect the trend over years, and over time it is likely that those of lower SES would have a larger reduction in HRQL than those who are wealthier.

Lower SES is associated with increased morbidity and mortality. A recent review of the association between SES and cardiovascular disease and/or cardiovascular risk factors, reported a clear inverse gradient in high income countries (*De Mestral & Stringhini, 2017*). Several studies have also identified a link between SES and type 2 diabetes (*Bird et al., 2015*; *Espelt et al., 2013*; *Hwang & Shon, 2014*). Early identification of modifiable risk factors and appropriate lifestyle changes can improve health outcomes and reduce the onset of chronic disease. However, if the health of at-risk middle-age men is not affecting their quality of life, they are less likely to meet a health professional. Health promotion efforts are also more likely to be acted upon by those of higher SES (*Adler & Newman, 2002*). Thus, our findings are of relevance as they underscore the importance of health promotion that focuses on disease prevention and identification of risk factors in men who *feel well*, particularly targeted to those of low SES.

The main limitations of our study were the secondary analyses of our clinical trial data, our relatively small number of participants ($n = 38$), and a selected group of individuals (overweight males living in a large urban centre), which may limit wider applicability of our findings. Particularly, since our cohort was relatively narrow and recruited from a single centre in New Zealand. Further, as all participants were volunteers in a clinical trial examining the effects of a nutritional supplement on metabolism, they are potentially more motivated than equivalent groups in the wider population. We also acknowledge that other factors that have been shown to affect HRQL (such as the education levels of the

individuals and their parents (*Ross et al., 2012*)) have not been controlled for, and could have affected study outcomes.

Nevertheless, our findings are likely to be robust as each participant underwent three assessments over a 30-week period, minimizing the potential effects of temporal variations in HRQL that individual participants might have experienced. In addition, the lack of associations in our study remained irrespective of whether SES was assessed using a measure of neighbourhood wealth (NZDep2006) or family wealth (capital value of residence). Lastly, we studied a relatively homogeneous group of men (mostly of New Zealand European ethnicity), which likely mitigated the confounding effects of ethnicity on study outcomes.

## CONCLUSION

Our findings show that SES per se was not associated with HRQL in middle-aged men who are overweight but otherwise healthy, an important and increasing group. While the equality of HRQL across the socioeconomic spectrum in this group would generally be considered favourable, overweight men of lower SES are known to have increased cardiometabolic risk. That this risk is not associated with impaired quality of life potentially reduces the likelihood that such men will to seek to improve their health. This finding underscores the fact that health promotion interventions aimed at improving the health of at-risk men should also aim to reach asymptomatic people who feel generally well.

### Funding
This study was supported by a TECHNZ grant (University of Auckland—UniS 30475.001) through the New Zealand Ministry of Science and Innovation. Martin de Bock was funded by the Joan Mary Reynolds Trust. The Paykel Trust has provided long-term funding for the Clinical Research Unit at the Liggins Institute (University of Auckland). The funders had no role in study design, data collection and analysis, decision to publish, or preparation of the manuscript.

### Grant Disclosures
The following grant information was disclosed by the authors:
New Zealand Ministry of Science and Innovation: UniS 30475.001.
Joan Mary Reynolds Trust.
The Paykel Trust.

### Competing Interests
José G.B. Derraik is an Academic Editor for PeerJ. The authors have no financial or non-financial conflicts of interest to disclose that may be relevant to this work.

### Author Contributions
- José G.B. Derraik conceived and designed the experiments, analyzed the data, prepared figures and/or tables, authored or reviewed drafts of the paper, approved the final draft.

- Benjamin B. Albert conceived and designed the experiments, prepared figures and/or tables, authored or reviewed drafts of the paper, approved the final draft.
- Martin de Bock conceived and designed the experiments, performed the experiments, authored or reviewed drafts of the paper, approved the final draft.
- Éadaoin M. Butler authored or reviewed drafts of the paper, approved the final draft, literature review.
- Paul L. Hofman conceived and designed the experiments, authored or reviewed drafts of the paper, approved the final draft.
- Wayne S. Cutfield conceived and designed the experiments, authored or reviewed drafts of the paper, approved the final draft.

### Human Ethics

The following information was supplied relating to ethical approvals (i.e., approving body and any reference numbers):

Ethical approval for this study was provided by the Northern Y Regional Ethics Committee from the New Zealand Ministry of Health (NTY/11/02/015).

### Data Availability

The raw data are provided as a Supplemental File.

### Supplemental Information

Supplemental information for this article can be found online at http://dx.doi.org/10.7717/peerj.5193#supplemental-information.

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
