# Peer review of "Socioeconomic status is not associated with health-related quality of life in a group of overweight middle-aged men"

_PeerJ, doi:10.7717/peerj.5193_

## Round 0.1 · original submission · Major Revisions

The reviewers suggest constructive comments to improve the manuscript. Statistics should be reconsidered in the light of the issues raised by reviewer 1.

·

Basic reporting

Introduction:
Lines 43 and 44: there are numerous studies that show the impact of socioeconomic status on the HRQL of adults,for example Alishiri et al and Ovayolu et al in their respective studies in 2008 and 2011 (Modern Rheumatology, 18(6), 601-608) (Clinical Rheumatology, 30(5), 655-664) indicated that precarious economic conditions were associated with a poor HRQOL. It is not necessary to provide evidence regarding the impact on adolescents when the focus is middle-aged adults

Lines 45-47: “Preliminary findings from our group indicate that among overweight middle-aged men, lower SES is associated with adverse metabolic outcomes (unpublished data)”. This assertion should have theoretical support based on studies already published.

Experimental design

Line 57: in the results refer that excluded individuals who did not complete the assessment of HRQL, should also be defined as an exclusion criterion.

Line 76: cite reference on the study of SF 36 validation in New Zealand
Line 87 it is necessary to detail the form of measurement and interpretation of the IPAQ. What sense did you have in the research, the evaluation of physical activity?

Lines 100-105: the demographic differences between the two groups to be compared should be evaluated using the t-student test or Mann Whitney U test. The ANOVA test and its non-parametric counterpart apply when there are three or more comparison groups, in this same sense the post hoc analyzes are also not necessary.

Why was not a descriptive and comparative analysis made of the other clinical variables (blood pressure, hypercholesterolemia) that were taken into account for the study population?

Validity of the findings

Results:
“Pearson's correlation coefficient or Spearman's rank correlation were initially run. Associations between NZDep2006 scores and SF-36v2 outcomes were examined using general linear mixed models based on repeated measures. Models adjusted for randomization sequence, timing of assessment, and on-going use o cholesterol-lowering and/or antihypertensive medications, as well as participant's age, BMI, and IPAQ score”. No results are presented that demonstrate the application of the statistical analysis proposed.

Line 136: As previously corrected, univariate analyzes do not allow the identification of associations.

Discussion:
Line 187 As previously recommended in the statistical analysis plan, the post hoc analyzes are not necessary for the comparison between the study groups,in this sense, it is not a limitation of the study, on the other hand I emphasize the recognition of the other limitations of the study for the purpose of new studies and interpretation of the results and conclusions of the study.

Reviewer 2 ·

Basic reporting

English language used was clear and unambiguous.

The motivation of the study was not strongly put across by authors.

Although the authors did mention the effect of SES on metabolic outcomes among overweight men using their preliminary findings, this evidence was not adequate enough to justify the reason for choosing middle-aged overweight men as the prime subject of focus and to study the impact of SES on HRQL among them.

Authors should cite past studies which have assessed this association in New Zealand or provide national statistics on the trend of obesity prevalence across gender over recent years to further stress the importance of studying the association between SES and HRQL among overweight men.

Experimental design

Authors should incorporate past literature and/or theories related to the study to identify gaps that underpins the aim and novelty of the study.

It is better for authors to clarify the use of a cohort study in this study as cohort study usually requires a longer period of time and interventions to assess changes in outcome variables.

Authors also did not take into account of control factors such as behavioural risk factors (smoking, alcohol consumption and healthy diet intake) which confounds the association between HRQL and SES and is also associated with being overweight. If information on this are not available, it would be better to acknowledge it as part of the limitation of the study.

Validity of the findings

Overall the findings of the study is quite interesting as it shows no association between SES and HRQL among overweight men. However this finding could be biased due to small sample size and its non-randomized sampling design. Hence the findings are not representative of the general population.

---

## Round 0.2 · Minor Revisions

According to the Reviewer 2, the authors could still improve the Discussion section by providing a speculative comment explaining the results obtained.

Reviewer 2 ·

Basic reporting

Article is written in clear and unambiguous English. Background of the study has improved from the previous version. Although the authors did mention that the study was not conducted in New Zealand, it would be better to acknowledge that such a study is limited internationally after an extensive review of the literature.

Experimental design

Research objectives and experimental design have improved from its previous version.

Validity of the findings

Findings are okay. Discussion of the study could be improved by elucidating more on the findings of the study as to why the authors have obtained a negative association.

---

## Round 0.3 · Minor Revisions

The manuscript has definitely improved, and reviewer 1 recommends to address just three minor issues (see also the annotated manuscript).

·

Basic reporting

In this section I consider that the suggestions made in the previous version are followed

Experimental design

For this section I only have three minor recommendations regarding the form of reference in the description of the collection instruments: SF-36 and IPAQ (see comments in the manuscript), in addition to the analysis plan proposed for the baseline comparison between the stratified groups specifically for the variable "Consumed any alcohol" (see tables 1 and 2)

Validity of the findings

Meet with the evaluation items of the journal

Additional comments

In general I think that the authors did a good job of revision and correction

---

## Round 0.4 · accepted · Accept

The authors have successfully addressed all the issues raised by reviewer.